# Apigenin Increases SHIP-1 Expression, Promotes Tumoricidal Macrophages and Anti-Tumor Immune Responses in Murine Pancreatic Cancer

**DOI:** 10.3390/cancers12123631

**Published:** 2020-12-04

**Authors:** Krystal Villalobos-Ayala, Ivannie Ortiz Rivera, Ciara Alvarez, Kazim Husain, DeVon DeLoach, Gerald Krystal, Margaret L. Hibbs, Kun Jiang, Tomar Ghansah

**Affiliations:** 1Department of Molecular Medicine, Morsani College of Medicine at the University of South Florida, Tampa, FL 33612, USA; krystalvilla@usf.edu (K.V.-A.); ivannieortiz@mail.usf.edu (I.O.R.); ciaraalvarez@usf.edu (C.A.); husaink@usf.edu (K.H.); 2Comparative Medicine at the University of South Florida, Tampa, FL 33612, USA; devon.deloach2@moffitt.org; 3The Terry Fox Laboratory, BC Cancer, Vancouver, BC V5Z 1L3, Canada; gkrystal@bccrc.ca; 4Department of Immunology and Pathology, Central Clinical School, Monash University, Melbourne 3004, Australia; margaret.hibbs@monash.edu; 5Anatomic Pathology Department, H. Lee Moffitt Cancer Center and Research Institute, Tampa, FL 33612, USA; kun.jiang@moffitt.org; 6Department of Immunology, H. Lee Moffitt Cancer Center, Tampa, FL 33612, USA

**Keywords:** SHIP-1, pancreatic cancer, apigenin, myeloid homeostasis, tumor immunity

## Abstract

**Simple Summary:**

Src Homology 2 (SH2) domain-containing Inositol 5’-Phosphatase-1 (SHIP-1) is an essential protein and the master regulator of myeloid cell development and function that impacts tumor immunity. We previously published that SHIP-1 regulates the expansion and function of immunosuppressive myeloid cells, which correlated with pancreatic cancer progression in mice. Here, we show that the bioflavonoid Apigenin restored SHIP-1 expression, significantly increased tumoricidal Tumor-Associated Macrophages (TAM) while significantly decreased immunosuppressive TAM percentages and improved anti-tumor immune responses in the tumor microenvironment using different pancreatic cancer models. Our research findings suggest that SHIP-1 may be a potential novel therapeutic target to promote the development of tumoricidal TAM that can assist in the treatment of pancreatic cancer.

**Abstract:**

Pancreatic cancer (PC) has an extremely poor prognosis due to the expansion of immunosuppressive myeloid-derived suppressor cells (MDSC) and tumor-associated macrophages (TAM) in the inflammatory tumor microenvironment (TME), which halts the recruitment of effector immune cells and renders immunotherapy ineffective. Thus, the identification of new molecular targets that can modulate the immunosuppressive TME is warranted for PC intervention. Src Homology-2 (SH2) domain-containing Inositol 5′-Phosphatase-1 (SHIP-1) is a lipid signaling protein and a regulator of myeloid cell development and function. Herein, we used the bioflavonoid apigenin (API) to reduce inflammation in different PC models. Wild type mice harboring heterotopic or orthotopic PC were treated with API, which induced SHIP-1 expression, reduced inflammatory tumor-derived factors (TDF), increased the proportion of tumoricidal macrophages and enhanced anti-tumor immune responses, resulting in a reduction in tumor burden compared to vehicle-treated PC mice. In contrast, SHIP-1-deficient mice exhibited an increased tumor burden and displayed augmented proportions of pro-tumor macrophages. These results provide further support for the importance of SHIP-1 expression in promoting pro-tumor macrophage development in the pancreatic TME. Our findings suggest that agents augmenting SHIP-1 expression may provide novel therapeutic options for the treatment of PC.

## 1. Introduction

Pancreatic cancer (PC) is one of the most aggressive and lethal cancers with a significantly higher mortality than other cancers [1]. Despite recent advances in cancer therapy, effective treatments for PC remain elusive. PC has heightened resistance to conventional treatments, such as immunotherapy and chemotherapy, in part, due to the inflammatory tumor microenvironment (TME) that contributes to the expansion of immunosuppressive myeloid-derived suppressor cells (MDSC) and Tumor-Associated Macrophages (TAM) [2,3]. These immunosuppressive cells specifically inhibit anti-tumor immune responses by halting the recruitment of effector immune cells to the TME, limiting the efficacy of current immunotherapies. In addition, MDSC and TAM interact with other immunosuppressive regulatory T cells (Tregs), immune cells, soluble factors, and other components in the stroma, which further exacerbates tumor progression, promotes metastasis and also chemoresistance [4,5,6]. Therefore, suppressing inflammation in the pancreatic TME may reduce the development of immunosuppressive MDSC and TAM and increase anti-tumor responses, thereby enhancing the efficacy of current therapies. 

A chronic inflammatory TME is established through the production of pancreatic tumor-derived factors (TDF) (i.e., GM-CSF, IL-6, IFN-γ, and MCP-1) that induce the expansion and recruitment of MDSC from the bone marrow (BM) [7,8]. These MDSC consist of immature myeloid cells, macrophages, granulocytes and dendritic cells (DC) that aggressively suppress anti-tumor immunity [9,10]. MDSC encompass two different subsets, monocytic (M-MDSC) and granulocytic (G-MDSC); both have different modes of action in suppressing tumor immunity [11]. Mouse and human M-MDSC are known to be more suppressive than G-MDSC [10]. In fact, the production of monocyte chemoattractant protein-1 (MCP-1), as well as other inflammatory chemokines produced by the tumor, induce mobilization of M-MDSC into the TME, where they differentiate into immunosuppressive TAM, which have the potential to further suppress anti-tumor immune responses in PC [8,12,13,14]. 

TAM are macrophages with plasticity that can change phenotype to become either immunogenic or immunosuppressive in response to cytokines, chemokines and other soluble factors in the TME [8,15]. TAM can be either M1-like macrophages (M1 TAM) that are tumoricidal, or M2-like macrophages (M2 TAM) that are pro-tumor; both can be found in the TME of mice and humans [15]. However, the pancreatic TME strongly favors M-MDSC polarization into M2 TAM, which facilitates tumor progression, metastasis and chemoresistance [8,15]. Importantly, the disruption of inflammatory cytokine and chemokine pathways has been shown to decrease M-MDSC recruitment into the TME, correlating with tumor regression in both pre-clinical models and patients with PC [8,16]. Thus, modulating MDSC expansion or their mobilization to the TME present potentially promising strategies to constrain MDSC–TAM-associated immunosuppression in PC. Hence, the identification of new potential molecular target(s) or pathways that can regulate the development, function or mobilization of MDSC–TAM may increase anti-tumor immune responses.

Src Homology-2 (SH2) domain-containing Inositol 5′-Phosphatase-1 (SHIP-1) is a 145 kDa protein that regulates the activity of immune cells including myeloid cells, macrophages and DC [17,18]. SHIP-1 expression is differentially and developmentally regulated in immune cells by external soluble factors such as cytokines and chemokines in the microenvironment [19]. A key study reported that SHIP-1 expression is important for repressing macrophage polarization, with alveolar, peritoneal and BM-derived macrophages from SHIP-1 knockout (KO) mice exhibiting an M2-like macrophage phenotype [20]. SHIP-1 negatively regulates phosphatidylinositol 3-kinase (PI3K) activity in immune cells [17,21,22]. As a result, SHIP-1 regulates the activity of numerous signaling pathways, including those involved in cell differentiation, proliferation, apoptosis, mobilization and function of myeloid immune cells [17,23]. Deficiency in SHIP-1 expression results in chronic myeloid leukemia in both humans and mice [24], consistent with studies reporting that SHIP-1 acts as a tumor suppressor preventing metastasis in a pre-clinical lung cancer model [25]. In previous studies, we reported a significant expansion of immunosuppressive macrophages in SHIP-1-deficient mice [26]. In addition, we also reported that mice with PC, SHIP-1 protein levels were significantly downregulated [21], and there were changes in the MDSC compartment that corresponded with an increase in tumor burden [21,27]. Thus, we have established that SHIP-1 is essential for maintaining myeloid homeostasis and function and shown that dysregulation of SHIP-1 can promote a pro-tumor microenvironment. Therefore, induction of SHIP-1 expression may augment tumoricidal TAM and improve anti-tumor immune responses in PC.

Recently, natural compounds known as bioflavonoids have been explored for their anti-tumor, anti-chemoresistance and anti-inflammatory properties against different cancers [28,29,30,31]. The bioflavonoid apigenin (API) has demonstrated potent anti-tumor activity and the ability to reduce chemoresistance to gemcitabine (one of the chemotherapy drugs used for PC) in human PC cell lines [32]. Amongst the bioactive flavonoids, API showed the most selective killing of cancer cells (i.e., rapidly dividing cells) while sparing normal cells [33]. In human PC cell lines, API induces apoptosis, cell-cycle arrest and inhibits DNA synthesis [28,34,35]. API has also been assessed in breast, prostate and PC pre-clinical animal models [36,37]. Our research group recently reported that API reduced tumor burden, improved anti-tumor immune responses and increased survival rates of mice bearing pancreatic tumors compared to vehicle-treated mice with PC [37]. 

In this study, we are the first to show that the bioflavonoid API reduces tumor-associated inflammatory factors, restores SHIP-1 expression and augments tumoricidal TAM, which corresponds with an increase in anti-tumor immune responses in different pre-clinical PC models. Specifically, we show that API treatment skews macrophages toward a tumoricidal M1-like phenotype in the TME of PC models. In addition, this skewing is associated with an increase in SHIP-1 expression and tumor regression in our PC models. These results suggest that SHIP-1 regulates the TME which can influence anti-tumor immune responses in PC. Therefore, SHIP-1 could be a potential therapeutic target for PC.

## 2. Results

### 2.1. API Reduces Tumor Burden in an Orthotopic Mouse Model of PC (OPC)

Bioflavonoids such as API have shown anti-tumor activity in experimental models of cancer [30,31,38]. We previously reported that API increased survival and reduced tumor burden in a heterotopic (H) mouse model of PC (HPC) [37]. To investigate if API had the same effect, we used a more clinically relevant orthotopic (O) mouse model of PC (OPC). Our results show that API treatment reduced tumor burden in OPC mice compared to vehicle-treated OPC mice (Figure 1A–C). Pancreatic tumors from OPC mice treated with the vehicle showed a significant increase in weight compared to the pancreas from control (CTRL) mice, while the pancreatic tumors from API-treated OPC mice weighed significantly less (Figure 1C), suggesting reduced pancreatic tumor burden. API treatment did not appear to cause toxicity as no significant differences in the body weights of the three cohorts of mice were observed (Figure 1D). Thus, using an OPC mouse model, which is more clinically applicable to humans with PC, we confirm that API exhibits anti-tumor activity.

### 2.2. API Suppresses Inflammation, Induces SHIP-1 Expression, and Reduces MDSC Expansion in OPC Mice

We previously reported that SHIP-1 expression was reduced due to the production of pancreatic TDF in HPC mice [21]. To examine this further, we investigated changes in pancreatic TDF and SHIP-1 expression, and correlated findings with proportions of MDSC and TAM in mice with HPC and OPC that had been treated with vehicle or API. In API-treated mice with HPC and OPC, a significant reduction in serum levels of IL-6, IFN-γ, MCP-1, TNF-α and IL-10 was observed compared to the levels seen in vehicle-treated HPC (Appendix A) and OPC mice (Figure 2A). 

Previously, we reported that SHIP-1 expression regulates MDSC homeostasis and function [21,27], so we considered the possibility that API might exert its anti-inflammatory effects, at least in part, by regulating SHIP-1 protein levels. To assess this, we analyzed the spleens of PC-bearing mice, which showed that the spleens from OPC mice were significantly enlarged (splenomegaly) compared to CTRL, while treatment with API reversed the splenomegaly (Figure 2B). We then performed flow cytometry on splenocytes, which showed that API treatment of HPC and OPC mice significantly reduced MDSC subsets percentages, G-MDSC (CD11b^+^Ly6C^+/−^Ly6G^+^) and M-MDSC (CD11b^+^Ly6G^−^Ly6C^+^), compared to vehicle-treated HPC mice (Appendix A) and OPC mice (Appendix A) (flow cytometry gating strategy defined in Appendix A). Accordingly, we wanted to investigate if the alteration of MDSC was the result of changes in SHIP-1 gene and protein expression. We found that SHIP-1 gene expression in splenocytes was reduced in PC-bearing mice and increased following treatment of mice with API (Appendix A). We then performed Western blot analysis using mouse anti-SHIP-1 (clone P1C1) [39], which confirmed that splenocytes from PC-bearing mice treated with API showed a significant increase in SHIP-1 protein (which appears as a doublet, 145 kDa and 135 kDa) compared to vehicle-treated mice (Appendix A and Figure 2C,D). Thus, in two different PC models, we have found that API treatment significantly reduces tumor-induced inflammation, leads to a restoration of SHIP-1 gene and protein levels and promotes MDSC homeostasis, which corresponds to a significant reduction in tumor burden [37]. 

As a control for this study, tumor-free C57BL/6N mice were treated with API (CTRL-API) and the vehicle. We observed no significant difference in mice and spleen weights from these tumor-free mice treated with API vs. vehicle (Appendix A). Flow cytometry results of splenocytes from these tumor-free mice showed no significant difference in the percentage of M-MDSC subset (Appendix A). Interestingly, our flow cytometry results showed a significant decrease in the percentage of splenic G-MDSC subset in tumor-free mice treated with API vs. vehicle (Appendix A). In addition, flow cytometry data showed no significant differences in the percentages of splenic CD8^+^ T cells. (Appendix A). Overall, API treatment showed no immunomodulatory effects on tumor-free mice regarding M-MDSC and CD8^+^ T cells compared to vehicle CTRL mice.

### 2.3. API Increases M1 TAM in the TME of OPC Mice 

We next investigated if API induced changes at the tumor site and, for this reason, we used flow cytometry to examine MDSC subsets as well as M1-like TAM (CD11b^+^Ly6C^+^Ly6G^−^F4/80^+^CD206^−^CD86^+^) and M2-like TAM (CD11b^+^Ly6C^+^Ly6G^−^F4/80^+^CD86^−^CD206^+^) in the TME of OPC mice (see the flow cytometry gating strategy defined in Appendix A). We found that API treatment significantly decreased the proportion of the G-MDSC subset in the TME but had no significant effect on the M-MDSC subset (Figure 3A,B). Very interestingly, API treatment significantly increased the percentages of M1-like TAM (anti-tumor) and decreased M2-like TAM (pro-tumor) in the TME (Figure 3C,D). In addition, we gated on Ly6C^low^ TAM in our flow analysis and it showed the same trend regarding a significant increase in M1-like TAM and a significant reduction in M2-like TAM percentages with API treatment (Appendix A). MDSC subsets are known to influence Treg expansion [9]. We previously reported that API reduces Treg percentages in our HPC model [37]. We now show by flow cytometry that API significantly decreases Treg (CD3^+^CD4^+^CD25^+^) percentages in the spleen and the TME of OPC mice (Appendix A). Thus, these results suggest that API increases SHIP-1 expression, leads to changes in MDSCs and Tregs and promotes the reprogramming of M2-like TAM into M1-like TAM in the TME of OPC mice compared to vehicle-treated OPC mice.

### 2.4. API Increases M1 TAM in Heterotopic KC-PC Mice 

Next, we asked if API’s immunomodulatory ability occurred in another clinically relevant model of PC (designated Kras^G12D^; Pdx1-Cre (KC)-HPC mice), making use of a novel pancreatic cancer cell line that mimics human PC due to its expression of mutated *Kras* [40]. Tumors were established in mice by heterotopic injection of UN-KC-6141 cells. Flow cytometry analysis of the spleen showed a significant increase in both G-MDSC and M-MDSC subsets percentages in tumor-bearing mice, which were significantly reduced by API treatment (Appendix A). As observed in our other PC models, Western blot and qPCR showed a significant reduction in SHIP-1 gene and protein expression in the spleen of KC-HPC compared to CTRL mice (Appendix A). API treatment of KC-HPC mice showed only an increase in SHIP-1 gene expression but showed a significant increase in SHIP-1 protein expression compared to vehicle-treated KC-HPC mice (Appendix A). Also, we observed a significant reduction in the tumor weights from KC-HPC-API mice compared to vehicle-treated KC-HPC mice (Figure 4A). Flow cytometry analysis of whole tumors revealed that API treatment of KC-HPC mice significantly decreased G-MDSC but did not alter M-MDSC percentages compared to vehicle-treated KC-HPC mice (Figure 4B,C). Next, when TAM subsets were assessed, API treatment of KC-HPC mice significantly increased M1-like TAM and significantly decreased M2-like TAM percentages in the TME compared to vehicle-treated KC-HPC mice (Figure 4D,E). In addition, with the KC-HPC model, we also gated on Ly6C^low^ TAM in our flow analysis and it showed the same trend regarding a significant increase in M1-like TAM and a significant reduction in M2-like TAM percentages with API treatment (Appendix A). Collectively, using three different PC models, we have shown that API can increase SHIP-1 expression and proportions of M1-like TAM in the TME.

### 2.5. M1 TAM are Reduced in Heterotopic PC SHIP^KO^ Mice 

SHIP-1 is important for the development and function of MDSC, macrophages and DC [17], thus impacting tumor immunity. We previously reported a downregulation of SHIP-1 expression in immunocompetent C57BL/6 mice with PC [21]. Thus, to examine SHIP-1′s role in PC development, we inoculated murine PC into SHIP^WT^ and SHIP^KO^ mice. We observed a significant increase in tumor burden in SHIP^KO^-HPC compared to SHIP^WT^-HPC mice inoculated with Panc02 cells (Figure 5A); however, no significant alterations in the proportions of G-MDSC and M-MDSC in the TME of SHIP^KO^-HPC vs. SHIP^WT^-HPC mice were found (Figure 5B,C). Nonetheless, due to the absence of SHIP-1 expression, SHIP^KO^-HPC had a significant increase in M2-like TAM and a significant decrease in M1-like TAM percentages in the TME compared to SHIP^WT^-HPC mice (Figure 5D,E). Additionally, we also gated on Ly6C^low^ TAM in our flow analysis and it showed the same trend regarding the percentages of M2-like TAM are significantly increased and M1-like TAM are significantly decreased in the TME of SHIP^KO^-HPC compared to SHIP^WT^-HPC mice (Appendix A). These results indicate the importance of SHIP-1 expression in myeloid cells for regulating the development of immunogenic M1-like TAM to control tumor progression in HPC mice. 

### 2.6. API Induces SHIP-1 Expression and Skews TAM to an M1 Phenotype in the TME of OPC Mice

We next investigated if API affects SHIP-1 expression in immune cells present in the TME of OPC mice. Flow cytometry analysis showed a significant increase in intracellular SHIP-1 expression in G-MDSC and M-MDSC from the TME of OPC-API compared to vehicle-treated OPC mice (Figure 6A,B). Moreover, qPCR and Western blotting showed a significant increase in SHIP-1 gene and protein expressions in the TME of OPC-API compared to vehicle-treated OPC mice (Figure 6C–E). Furthermore, our Western blotting results show minimal SHIP-1 protein levels in the CTRL pancreas (Appendix A). We also assessed the protein expressions of inducible nitric oxide synthase (iNOS), chitinase 3-like 3 (known as Ym-1) and arginase-1 (Arg-1) in the TME of OPC-API and OPC mice to distinguish M1-like TAM from M2-like TAM [10,20]. We found a significant increase in iNOS (M1-like TAM) and a significant decrease in Ym-1 (M2-like TAM) expressions in OPC-API compared to vehicle-treated OPC mice (Figure 6D,E). However, we did not observe a significant difference in Arg-1 expression, another M2 TAM marker (Figure 6D,E). These results suggest that immune cells that express SHIP-1 can mobilize to the TME and that API upregulates SHIP-1, promoting the development of M1-like TAM in the TME of OPC-API mice. 

### 2.7. API Improves Anti-Tumor Immune Responses in the TME of OPC Mice

The pancreatic TME prevents mobilization of effector immune cells to the tumor, which limits the efficacy of treatments [3,6,41]. Therefore, we investigated the effects of API on the mobilization of CD3^+^ tumor infiltrating lymphocytes (TILs) and anti-tumor activity in the TME of OPC mice. Immunohistochemistry analysis showed a significant increase in the numbers of CD3^+^ TILs in pancreas/tumors from OPC-API compared to vehicle-treated OPC mice (Figure 7A,B). Histopathology of pancreas/tumors from OPC-API mice revealed necrosis and an increase in normal stromal tissue compared to vehicle-treated OPC mice (Figure 7C). In addition, Western blotting showed a significant reduction in the expression of Bcl-2 in the TME of API-treated OPC mice compared to vehicle-treated OPC mice (Appendix A). Flow cytometry analysis demonstrated a significant increase in the intra-tumoral CD8^+^ T cells (CD3^+^CD8^+^) percentages, as well as a significant increase in its production of IFN-γ when stimulated in vitro, from the TME of OPC-API compared to vehicle-treated OPC mice (Figure 7D–F). Furthermore, Western blotting results showed a significant increase in granzyme B protein expression but not in perforin expression (Figure 7G,H). Interestingly, qPCR results showed a significant increase in both perforin and granzyme B gene expression in the TME of OPC-API compared to OPC mice (Appendix A). Additionally, our immunofluorescence data showed a significant increase in perforin and an increase trend in granzyme B in the TME of OPC-API vs. OPC mice (Appendix A). These results strongly suggest that API improves the mobilization of TILs and enhances anti-tumor activity in the pancreatic TME of OPC mice.

## 3. Discussion

One of the major barriers for PC treatment is the expansion and accumulation of immunosuppressive MDSC and TAM that inhibit effector immune cells mobilizing to the TME to eradicate the tumor [3,6]. Thus, identifying new molecular targets that regulate MDSC and TAM development is urgently needed to reduce immunosuppression and to improve the efficacy of current PC treatments. In this study, we are the first to show that SHIP-1 gene and protein expression is dampened in the setting of PC. In addition, we show that the anti-inflammatory drug API reduced inflammatory cytokines and chemokines in experimental PC models, which correlated with an increase in SHIP-1 expression. This restoration in SHIP-1 expression by API treatment significantly promoted the development of anti-tumor M1-like TAM in the TME of these pre-clinical models. More importantly, API treatment increased the mobilization of effector CD8^+^ T cells and augmented their anti-tumor activity in the TME of OPC mice. In addition, we show that pancreatic tumors grow more rapidly in SHIP^KO^ mice and their TAM are skewed toward a pro-tumor phenotype (M2-like TAM). This suggests that SHIP-1 is an important regulator of macrophage skewing in the TME of PC. Our results support the notion that SHIP-1 controls the plasticity of macrophages, with its enhanced expression promoting immunogenic M1-like TAM over immunosuppressive M2-like TAM in the pancreatic TME. Therefore, amplification of SHIP-1 expression may be a novel means by which macrophages could be polarized towards an anti-tumor phenotype in the pancreatic TME for the treatment of PC. 

SHIP-1 expression is important for appropriate regulation of hematopoiesis and the maturation of myeloid cells into granulocytes, macrophages, and DC (Figure 8). Loss of SHIP-1 expression impacts hematopoietic stem cells (HSC) and exacerbates altered myelopoiesis, which leads to the development of immunosuppressive MDSCs, Tregs, and pro-tumor M2-like TAM in the pancreatic TME, thus facilitating PC tumor progression (Figure 8). In this study, we showed that API treatment reduces pancreatic TDFs such as IL-6, IFN-γ, TNF-α and MCP-1, which in turn, restores SHIP-1 expression. This likely leads to an adjustment of myelopoiesis and the development of tumoricidal M1-like TAM in PC models (Figure 8). API is a bioflavonoid known to have multiple targets such as casein kinase II (CK2), microRNAs (miRNAs), cytokine and chemokine signaling pathways, and it is an inhibitor of proteasomal degradation [37,42]. Currently, we do not know the exact targets that induce transcriptional suppression of SHIP-1 in our PC models and we are performing additional experiments to identify these mediators. 

G-MDSC can be transformed into Tumor-Associated Neutrophils (TAN) in the TME, but this phenomenon has been poorly investigated [43]. These TAN consist of N1-like TAN (N1 TAN) that are tumoricidal and N2-like TAN (N2 TAN) that have pro-tumor activity [43]. These TAN are recruited to the TME by inflammatory cytokines and chemokines and can only be distinguished due to their activation and cytokine production [43]. We observed that API treatment caused a significant reduction in the proportion of G-MDSC but no difference in M-MDSC in the TME of OPC and KC-HPC models. Interestingly, we observed a significant reduction in splenic G-MDSC percentages in CTRL-API (tumor-free) compared to vehicle-treated CTRL mice. It appears that API reduces G-MDSC percentages which is a new finding we are currently investigating. The phenotypic markers that distinguish N1 and N2 TAN in the TME of PC mice are unclear but may also include neutrophil chemokine receptors such as CXCR2 [43]. Our studies implicate SHIP-1 expression as a contributor to G-MDSC homeostasis and the regulation of TAN (N1 vs. N2) in the TME that may influence PC progression and treatment resistance. 

M-MDSC that mobilize to the TME develop into TAM [11]. However, we observed no differences in the proportion of M-MDSC vs. G-MDSC in the TME of API-treated OPC or KC-HPC mice, despite significant differences in M-MDSC and G-MDSC subsets in the spleen of HPC, OPC and KC-HPC mice treated with API. It has been reported that inflammatory monocytic-derived dendritic cells (inf-MoDC) are present in the TME, where they are phenotypically and physiologically distinct from M1 and M2 TAM [10,14]. It is known that SHIP-1 is important for DC development and function [18]. Therefore, we speculate that inf-MoDC are present and may have a dominant role in the TME of our OPC and KC-HPC models. This may explain why we did not observe any difference in a proportion of the M-MDSC subset in our models in the presence or absence of API. We have yet to fully immunophenotype inf-MoDC according to the current literature [14] or examine responses to API. 

SHIP-1 expression is known to influence the development of immunosuppressive Treg and immunosuppressive TH17 immune cells in association with MDSC [9,21,44]. However, we did not evaluate TH17 percentages in our PC models. Yet, the mechanisms by which Treg and TH17 interplay with MDSC in the pancreatic TME treated with and without API warrants further investigation. 

It is important to discuss Src homology 2 (SH-2) domain-containing tyrosine phosphatase-1 and -2 (SHP-1 and SHP-2) and their influence on PC progression [45,46]. SHP-1 acts as a tumor suppressor that potentially regulates immune checkpoint molecule expression, which can impact TIL mobilization into the TME of PC [45]. The overexpression of SHP-2 is considered to be biomarker and sign of poor prognosis for PC [46]. Therefore, SHP-2 acts as an oncogene that is important for *Kras* activation in human PC, which impacts signaling events that can control PC progression [47,48]. We reported in this study that API treatment of our OPC mice resulted in an increase in the mobilization of TILs into the TME that corresponded with a reduction in tumor burden. Moreover, we reported that API reduced PC induced inflammatory factors that are known to be potentially governed by downstream *Kras* signaling events (i.e., Janus Kinase/Signal Transducer and Activator of Transcription (JAK/STAT) cytokine-dependent signaling pathways). However, the mechanisms by which API regulates SHP-1/SHP-2 activity and signaling impacts PC outcomes warrant further investigation. 

There are limitations in this study. First, a normal pancreas lacks infiltrating immune cells [49], therefore we were not able to evaluate its immune cell compartment even following enzymatic tissue digestion, in comparison to pancreatic tumors that are readily infiltrated by MDSCs, TAM and CD8^+^ T cells. Secondly, we were not able to yield enough pancreas protein lysates from CTRL mice (no tumor), which coincided with minimal SHIP-1 expression observed in our results from Western blot. Thirdly, API as a single agent therapy may be limited, however, we used API as an anti-inflammatory drug and as a tool to reduce tumor-induced inflammation. We found that API increased SHIP-1 expression, skewed macrophages to an anti-tumor phenotype and delayed tumor progression in our PC models. Therefore, in a clinical setting API may be used as an adjuvant with other PC therapeutic regimen. More importantly, we show here and previously published that API therapy (25 mg/kg) has no apparent side effects in our established PC models [37], which is consistent with its non-toxic properties in other pre-clinical models of cancer [50,51]. In fact, there are several other pre-clinical studies using higher dosages (i.e., 300 mg/kg and 50 µg/day) and longer durations of API treatments (up to 20 weeks) [52,53,54]. In this study, we use API as a tool to increase SHIP-1 expression, the percentages of tumoricidal macrophages (M1-like TAM) and anti-tumor immunity in pre-clinical PC models. Taken together, our results suggest that SHIP-1 may be a potential and novel therapeutic target to be used as an interventional strategy for treating human pancreatic cancer.

## 4. Materials and Methods

### 4.1. Pancreatic Cancer Cell Lines

The murine Panc02 adenocarcinoma cell line originated from C57BL/6 mice [55]. The murine UN-KC-6141 cell line was derived from a C57BL/6 mouse bearing a Kras^G12D^; Pdx1-Cre (KC) pancreatic tumor [40]. The Panc02 and UN-KC-6141 cell lines were maintained in Roswell Park Memorial Institute (RPMI) 1640 and Dulbecco’s Modified Eagle’s Medium (DMEM) (+4.5g/L D-Glucose, L-Glutamine) (ThermoFisher Scientific, Waltham, MA, USA), respectively, supplemented with 10% fetal bovine serum (FBS) (HyClone), 100 U/mL penicillin and 100 μg/mL streptomycin (Gibco) at 37 °C in 5% CO_2_. Cultured cells were tested and found to be negative for mycoplasma and viral contamination.

### 4.2. Bioflavonoid 

Apigenin (API) (4′,5,7-Trihydroxyflavone, 5,7-Dihydroxy-2-(4-hydroxyphenyl)-4-benzopyrone) was purchased from Sigma-Aldrich (St. Louis, MO, USA) and diluted in dimethyl sulfoxide (DMSO) according to the manufacturer’s instructions.

### 4.3. Pancreatic Cancer Murine Models

All female C57BL/6N mice were purchased from Envigo (Indianapolis, IN, USA) and maintained in a pathogen-free animal facility for one week before the start of experiments. Mice (6–8 weeks of age) were subcutaneously (SC) injected in the lower ventral abdomen with 1.5 × 10^5^ murine Panc02 or 5 × 10^6^ murine UN-KC-6141 cells, in phosphate buffer solution (PBS) suspension, generating heterotopic pancreatic cancer (HPC) or KC-HPC mice, respectively. Control (CTRL) mice were SC injected with sterile PBS. Treatments were started once palpable tumors were apparent. Cohorts of HPC and KC-HPC mice received either 100 µL of sterile PBS (vehicle) or doses of 25 mg/kg of API (HPC-API and KC-HPC-API) administered by intraperitoneal (IP) injections three times per week until the end of the study [37,50]. CTRL mice received 100 µL of sterile PBS (vehicle) via IP injections three times per week until the end of the study [37]. The endpoint of the study was reached between 21 and 28 days post-injection for the HPC model and 16–17 days post-injection for the KC-HPC model. 

In another parallel experiment, C57BL/6N (6–8 weeks of age) mice were SC injected with sterile PBS. These CTRL mice were then treated with doses of 25 mg/kg of API (CTRL-API tumor-free mice) administered by IP injections three times per week until the end of the study. The endpoint of the study was reached between 16 and 17 days post-injection.

Female C57BL/6 SHIP^KO^ and SHIP^WT^ mice were originally obtained from the Krystal Research Group [56], and for more recent experiments, were obtained from the Hibbs Lab [57]. These mice (4–6 weeks of age) were SC injected with 1.5 × 10^5^ Panc02 cells, in PBS suspension, in the lower ventral abdomen generating SHIP^KO^-HPC and SHIP^WT^-HPC mice. The endpoint of the study was reached between 14 and 15 days post-injection. 

To generate orthotopic models, C57BL/6N mice (6–8 weeks of age) were anesthetized with 1.5–3% isoflurane and injected with sterile PBS (CTRL) or 1.25 × 10^4^ Panc02 cells (OPC), in PBS suspension, into the neck of the pancreas via laparotomy [58]. Post-surgery ultrasound imaging (Vevo 2100) was performed to confirm the presences of pancreatic tumors before the start of API treatments. A cohort of OPC mice received either 100 µL of sterile PBS (vehicle) or doses of 25 mg/kg of API (OPC-API) administered by IP injections three times per week until the end of the study. CTRL mice received 100 µL of sterile PBS (vehicle) via IP injections three times per week until the end of the study. The endpoint of the study was reached between 16 and 20 days post-surgery. Mice were weighed at the endpoint of this study.

All mice were humanely euthanized using CO_2_ and cervical dislocation, according to the University of South Florida (USF) Institutional Animal Care and Use Committee (IACUC) guidelines. The endpoint of all our PC models were reached at different time points due to the use of mice with different genetic backgrounds and different cell lines. This in turn caused rapid tumor necrosis and/or ascites in our mice. To account for this we closely monitored the mice and used the same USF IACUC guidelines for endpoint criteria for all strains and cell lines which included: severe symptoms of tumor necrosis, hunched posture (kyphosis), abnormal breathing patterns, isolation from cage mates, failure to groom, inability to eat or drink, vocalization, pale extremities, and/or ascites. When any of these symptoms were observed in our mice with PC, they were euthanized along with their CTRL counterparts. Lymphoid tissues (i.e., peripheral blood (PB), tumors, spleens) were harvested, weighed, and processed for biochemical assays in this study. The USF IACUC approved protocols IS00004664 and IS00004665 abides by the Guide for the Care and Use of Laboratory Animals.

### 4.4. Ultrasound Imaging 

High-resolution ultrasound imaging was done using a Visual Sonics Vevo 2100 Imaging system performed at Moffitt Small Animal Imaging Lab (SAIL) Core. Orthotopic PC mice were anesthetized with 1.5–3% isoflurane before each imaging session and maintained at the same flow rate. Mice were imaged weekly to detect the pancreas, pancreatic tumors, and tumor area. Vevo Lab (Visual Sonics, Toronto, ON, Canada) software was used to analyze the images, analyzed at Moffitt SAIL Core.

### 4.5. Flow Cytometry

Splenocytes were treated with Red Blood Cell lysis buffer (eBioscience, San Diego, CA, USA), according to the manufacturer’s instructions. All pancreatic tumors were digested with Collagenase, type IV (Sigma-Aldrich), DNase, type IV (Sigma-Aldrich) and Hyaluronidase, Type V (Sigma-Aldrich) in Hank’s Balanced Salt Solution (without CaCl_2_ or MgCl_2_) for 1 h [59]. Single cell suspensions were counted and then resuspended in 3% FBS/PBS and surface stained with fluorescent antibodies against murine T cell and myeloid cell surface markers, as previously described [60]. MDSC–TAM subsets were detected with anti-mouse CD11b-APC, anti-mouse Ly6C-PE/Cy7, anti-mouse Ly6G-PerCP, anti-mouse F4/80-BV650, anti-mouse CD86-AlexaFluor700 and anti-mouse CD206-FITC (Biolegend, San Diego, CA, USA). T cells were detected with anti-mouse CD3-FITC, anti-mouse CD8-PercP/Cy5.5, anti-mouse CD4-APC/Cy7 and anti-mouse CD25-BV605 (Biolegend). All samples were then fixed with 2% paraformaldehyde (PFA) for 15 min on ice. OPC and OPC-API tumor samples were intracellularly stained with anti-mouse SHIP-1-PE (Clone: P1C1-A5; Biolegend), along with isotype CTRL, using a Cytofix/Cytoperm kit (BD Bioscience, San Jose, CA, USA) according to the manufacturer’s protocol. Acquisition of samples was performed using a flow cytometer BD LSRII (BD Biosciences Immunocytometry Systems). FlowJo software (BD Bioscience) was used for data analysis. Flow cytometry gating strategy for MDSC and TAM subsets are defined in Appendix A. MDSC subsets were defined as, G-MDSC (CD11b^+^Ly6C^+/−^Ly6G^+^) and M-MDSC (CD11b^+^Ly6G^−^Ly6C^+^), and TAM subsets were defined as M1-like TAM (CD11b^+^Ly6C^+^Ly6G^−^F4/80^+^CD206^−^CD86^+^) and M2-like TAM (CD11b^+^Ly6C^+^Ly6G^−^F4/80^+^CD86^−^CD206^+^).

### 4.6. T Cell Activation Assay

Single-cell suspensions of digested tumors from OPC and OPC-API mice (1 × 10^6^ cells) were plated onto a 96 well plate coated with purified anti-mouse CD3 (10µg/mL) (Biolegend). Soluble purified anti-mouse CD28 (2µg/mL) (Biolegend) in RPMI 1640 media (10% FBS, 100 U/mL penicillin and 100 μg/mL streptomycin) was added to the assay wells and then incubated at 37 °C in 5% CO_2_. After two days of incubation, Golgi Plug, containing brefeldin A, (1 ug/mL) (BD Bioscience) was added to each assay well and incubated for 4 hrs. After incubation, cells were surface stained with fluorescent antibodies including anti-mouse CD3-FITC (Biolegend) and anti-mouse CD8-PercP/Cy5.5 (Biolegend). Samples were then intracellularly stained with anti-mouse IFN-γ-PE (eBioscience), along with isotype CTRL, using a Cytofix/Cytoperm kit (BD Bioscience), according to the manufacturer’s protocol. Acquisition of activated CD8^+^ T cell samples were performed using a flow cytometer BD LSRII (BD Biosciences Immunocytometry Systems). FlowJo software (BD Bioscience) was used for data analysis.

### 4.7. Western Blot

Single-cell suspensions of splenocytes and digested pancreatic tumors were lysed using modified radioimmunoprecipitation assay (RIPA) Buffer (Millipore, Burlington, MA, USA) supplemented with Na_3_OV_4_ and protease inhibitor cocktail (Sigma-Aldrich) [37]. Protein concentrations were determined by the BCA Protein Assay Kit (Thermo Fisher Scientific). Approximately 30 µg of splenocytes and tumor protein lysates, were loaded and resolved using NuPAGE 4–12% Bis-Tris polyacrylamide gels and transferred to nitrocellulose membranes, using the XCell SureLock (Invitrogen, Waltham, MA, USA). The nitrocellulose membranes were blocked using 5% non-fat milk in PBS/0.1% Tween-20 and then probed with anti-SHIP-1 (Santa Cruz Biotechnology, Dallas, TX, USA), anti-iNOS, anti-Arginase-1, anti-Granzyme-B, anti-Perforin (Cell Signaling Technology, Danvers, MA, USA) and anti-Bcl-2 (Santa Cruz Biotechnology), all at a dilution of 1:1000. Anti-Ym-1-HRP conjugated (Biolegend) was used at a dilution of 1:20,000. As an internal control for equal protein loading, all blots were re-probed with anti-β-actin, HRP-conjugated (Sigma-Aldrich), at a dilution of 1:20,000. Primary antibodies were detected using the respective secondary anti-mouse or anti-rabbit IgG, HRP-conjugated antibodies (Cell Signaling Technology), at a dilution of 1:1000. Super Signal West Pico and Femto Chemiluminescent Substrates (Thermo Fisher Scientific) were used to identify the secondary antibodies. Nitrocellulose membranes were imaged using the Bio-Rad ChemiDoc XRS Imaging System. Quantification of band intensities was performed using Image J.

### 4.8. Quantitative PCR (qPCR)

Total RNA was extracted from single-cell suspensions of splenocytes and digested pancreatic tumors using RNeasy Mini Kit (Qiagen, Germantown, MD, USA), according to manufacturer’s instructions. RNA was quantified using a NanoDrop 1000 (ThermoFisher Scientific) and then normalized. RT-PCR was performed using a High Capacity cDNA RT Kit with RNase Inhibitor (Applied Biosystems, Waltham, MA, USA) according to manufacturer’s instructions. mRNA levels for SHIP-1, perforin and granzyme B were detected by qPCR using iQ SYBR Green Supermix (Bio-Rad, Hercules, CA, USA) with an Eppendorf Master cycler realplex 4. The following primers were designed and purchased from Integrated DNA Technologies: SHIP-1 forward, 5′-CCA GGG CAA GAT GAG GGA GA-3′, SHIP-1 reverse, 5′-GGA CCT CGG TTG GCA ATG TA-3′, granzyme B forward, 5′-ATC AAG GAT CAG CAG CCT GA-3′, granzyme B reverse, 5′-TGA TGT CAT TGG AGA ATG TCT-3′ [61], perforin forward, 5′-TCA TCA TCC CAG CCG TAG T-3′, perforin reverse, 5′-ATT CAT GCC AGT GTG AGT GC-3′ [62], and as the internal control and reference gene, GAPDH forward, 5′-TGA TGG CGT GGA CAG TGG TCA TAA-3′, GAPDH reverse, 5′-CAT GTT TGT GAT GGG CGT GAA CCA. SHIP-1 and GAPDH primers were used at the following conditions: 95 °C for 3 min followed by 40 cycles of 95 °C for 15 s and 60 °C for 1 min. Granzyme B and perforin primers were used at the following conditions: 95 °C for 3 min followed by 40 cycles of 95 °C for 15 s and 55 °C for 1 min. Each sample was assayed in triplicate. The relative SHIP-1, perforin and granzyme B mRNA expression was calculated using the Comparative 2^−∆∆Ct^ method.

### 4.9. Cytometric Bead Array

Cardiac PB was collected from all HPC and OPC mice in this study. Serum was quickly processed from PB, flash frozen and stored at −80 °C. Serum was thawed to detect inflammatory factors (cytokines and chemokines) using mouse Cytometric Bead Array Inflammation Kit (BD Bioscience), according to the manufacturer’s instructions. Flow cytometry was performed using a BD FACS Canto II. FCAP Array software was used for data analysis.

### 4.10. Immunohistochemistry Staining

Formalin-fixed (10%, 24 h) OPC and OPC-API pancreatic tissue slices, including tumors, were used and sectioned to detect TILs via immunohistochemistry using rabbit anti-mouse-CD3 (Spring Biosciences, Pleasanton, CA, USA), at a dilution of 1:100 for 32 min, and OmniMap anti-rabbit secondary antibody (Ventana, Tuscon, AZ, USA) for 8 min. CD3^+^ TILs were detected using a ChromoMap Kit used on a Discovery XT automated system (Ventana), following manufacturer’s instructions. Staining was performed at Moffitt Cancer Center (Tampa, FL, USA). Moffitt Cancer Center Pathologist took high magnification (600×) and high-resolution microphotographs of OPC and OPC-API tumor slides (blinded) using an Olympus Model BX43 microscope. Individual CD3^+^ TILs (denoted by cells enveloped in the brown detection color) were counted as the pathologist instructed. Non-specific binding was not detected using appropriate controls.

### 4.11. Indirect Immunofluorescence

Single-cell suspensions of digested tumors from OPC and OPC-API mice (3.5 × 10^4^ cells/well) were seeded onto polytetrafluoroethylene slide wells (ThermoFisher Scientific). The cells were blocked with 1.5% Bovine serum albumin in PBS for 30 min, then fixated with 4% PFA for 15 min. The cells were permeabilized with 0.1% Triton X-100 in PBS for 15 min for intracellular staining. The cells were primary immunostained with purified rabbit anti-mouse granzyme B (Cell Signaling Technology) at 4 °C overnight, at a dilution of 1:100. The cells were secondary immunostained with goat anti-rabbit conjugated to Alexa fluor 488 (Abcam, Cambridge, MA, USA) for 1-h at room temperature (RT), at a dilution of 1:200. The cells were direct immunostained with rat anti-mouse perforin conjugated to Alexa fluor 546 (Santa Cruz Biotechnology) for 1 h at RT, at a dilution of 1:200. A drop of ProLong GOLD antifade reagent with 4′,6-diamidino-2-phenylindole (DAPI; ThermoFisher Scientific) was used for each assay wells and topped with a coverslip. The slides were observed using an Olympus BX53 digital fluorescence microscope at the USF microscopy core. The images were analyzed using ImageJ software to determine relative fluorescence magnitude.

### 4.12. Hematoxylin and Eosin Staining 

The hematoxylin and eosin (H&E) sections of OPC and OPC-API pancreatic tissue slices including tumor were prepared, examined, and interpreted for necrosis of pancreatic tumor cells, viable stroma and viable tumor cells. High magnification (100×) and high-resolution microphotographs of OPC and OPC-API tumor slides were taken using an Olympus Model BX43 microscope. This was performed at the Pathology Center, Moffitt Cancer Center (Tampa, FL, USA).

### 4.13. Statistical Analysis

All in vivo and in vitro experimental results are represented of the mean ± Standard Deviation (S.D.) of at least three independent biological replicates analyzed by unpaired two-tailed *t* tests. All in vivo PC models were independently repeated at least twice (*n* = 3–6 per each experimental group). Representative quantification of normalized densitometric ratios of Western blot (divided by internal CTRL protein) data was performed using Image J. Differences were considered significant at *p* < 0.05. All statistical analyses were performed using Prism 8 Software (GraphPad, San Diego, CA, USA).

## 5. Conclusions

This study highlights the importance of SHIP-1 as a potential tumor suppressor that regulates immunosuppressive MDSC and TAM development and function, which impact anti-tumor immune responses in PC pre-clinical models. Our results strongly suggest that the dietary bioflavonoid API may be therapeutically beneficial in restoring SHIP-1 expression via repressing inflammatory pancreatic tumor-derived factors. Therefore, API therapy increases SHIP-1 expression, restoring normal myelopoiesis which, in turn, reduces immunosuppressive M2-like TAM and converts them into immunogenic M1-like TAM in the TME, thus enhancing anti-tumor immune responses (i.e., CD8^+^ T cells) and leading to pancreatic tumor regression (Figure 8). The results from this study suggest that augmenting SHIP-1 expression may be a novel means to promote tumoricidal macrophages for the treatment of pancreatic cancer.

## Figures and Tables

**Figure 1 cancers-12-03631-f001:**
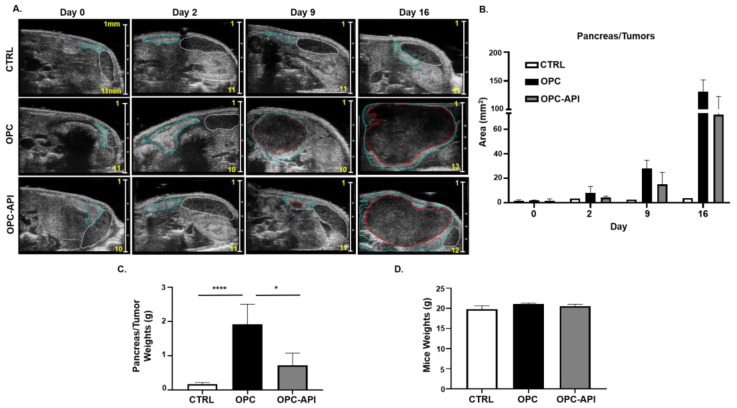
Apigenin (API) Reduces Tumor Burden in OPC Mice. (**A**) 2100 Vevo Ultrasound imaging of control (CTRL), orthotopic (O) mouse model of pancreatic cancer (PC) (OPC) treated with vehicle and OPC treated with API mice was performed once a week to detect the pancreas (outlined in blue), tumor (outlined in red) and spleen (outlined in white). Numbers on scale bar represents millimeters. (**B**) Representative quantification of pancreas/tumor area of CTRL, OPC and OPC-API-treated mice. (**C**) Weights of the pancreas/tumors of CTRL, OPC and OPC-API mice at the end of the study. (**D**) Mice weights at the end of the study. Data are represented as the mean ± S.D. of CTRL (*n* = 3–6), OPC (*n* = 3–4) and OPC-API (*n* = 3–4) mice. * *p* < 0.05; **** *p* < 0.0001 (by two-tailed *t* test).

**Figure 2 cancers-12-03631-f002:**
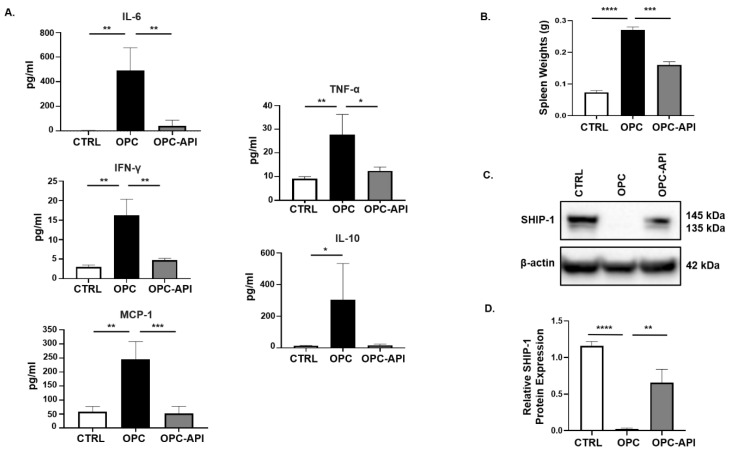
API Reduces Inflammation and Induces SHIP-1 Expression in OPC Mice. (**A**) Cytokine and chemokine profiles from the serum of CTRL, OPC and OPC-API mice as measured by Cytometric Bead Array and flow cytometry. (**B**) Quantification of spleen weights of CTRL, OPC and OPC-API mice. (**C**,**D**) Western blot analysis and quantification of normalized densitometry ratios of Src Homology-2 (SH2) domain-containing Inositol 5′-Phosphatase-1 (SHIP-1) protein in whole-splenocyte lysates from CTRL, OPC and OPC-API mice. Appendix A shows the uncropped images. Data are represented as the mean ± S.D. of CTRL (*n* = 3–4), OPC (*n* = 3–4) and OPC-API (*n* = 3–5) mice. * *p* < 0.05; ** *p* < 0.01; *** *p* < 0.001; **** *p* < 0.0001 (by two-tailed *t* test).

**Figure 3 cancers-12-03631-f003:**
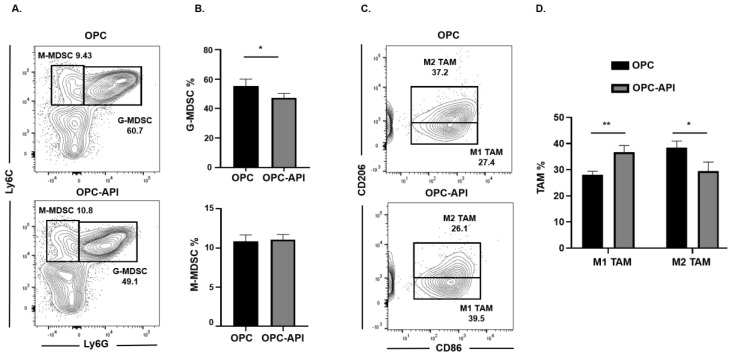
API Modulates Myeloid-Derived Suppressor Cell (MDSC) and Tumor-Associated Macrophage (TAM) Subsets in the Tumor Microenvironment (TME) of OPC Mice. (**A**,**B**) Flow cytometry analysis and representative quantification of MDSC subsets, G-MDSC (CD11b^+^Ly6C^+/−^Ly6G^+^) and M-MDSC (CD11b^+^Ly6G^−^Ly6C^+^), in whole pancreatic tumors from OPC and OPC-API-treated mice. (**C**,**D**) Flow cytometry analysis and quantification of TAM subsets, M1-like TAM (CD11b^+^Ly6C^+^Ly6G^−^F4/80^+^CD206^−^CD86^+^) and M2-like TAM (CD11b^+^Ly6C^+^Ly6G^−^F4/80^+^CD86^−^CD206^+^) in whole pancreatic tumors from OPC and OPC-API mice. Data are represented as the mean ± S.D. of OPC (*n* = 3) and OPC-API (*n* = 3–4) mice. * *p* < 0.05; ** *p* < 0.01 (by two-tailed *t* test).

**Figure 4 cancers-12-03631-f004:**
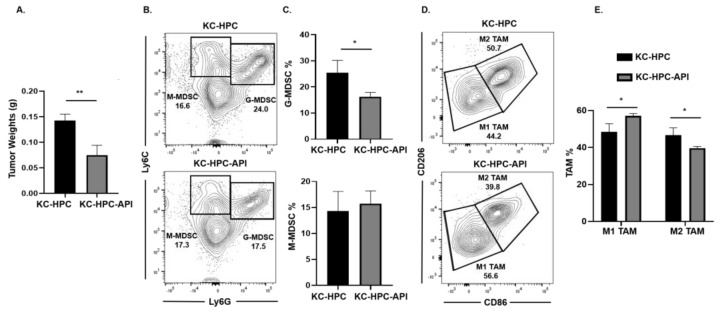
API Increases M1-like TAM in Heterotopic Kras^G12D^; Pdx1-Cre (KC)-PC Mice. (**A**) Weights of HPC tumors from KC-HPC and KC-HPC-API mice at the endpoint of the study. (**B**,**C**) Flow cytometry analysis and representative quantification of MDSC subsets, G-MDSC (CD11b^+^Ly6C^+/−^Ly6G^+^) and M-MDSC (CD11b^+^Ly6G^−^Ly6C^+^), using whole tumors of KC-HPC and KC-HPC-API mice. (**D**,**E**) Flow cytometry analysis and quantification of TAM subsets, M1-like TAM (CD11b^+^Ly6C^+^Ly6G^−^F4/80^+^CD206^−^CD86^+^) and M2-like TAM (CD11b^+^Ly6C^+^Ly6G^−^F4/80^+^CD86^−^CD206^+^), using whole tumors from KC-HPC and KC-HPC-API mice. Data are represented as the mean ± S.D. of KC-HPC (*n* = 3–4) and KC-HPC-API (*n* = 3–4) mice. * *p* < 0.05; ** *p* < 0.01 (by two-tailed *t* test).

**Figure 5 cancers-12-03631-f005:**
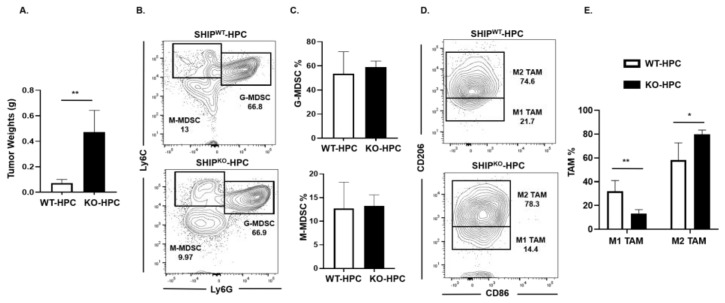
SHIP-1 Expression Controls the Development of M1-like TAM in HPC Mice. (**A**) Weights of HPC tumors from SHIP^WT^ and SHIP^KO^ mice at the endpoint of the study. (**B**,**C**) Flow cytometry analysis and representative quantification of MDSC subsets, G-MDSC (CD11b^+^Ly6C^+/−^Ly6G^+^) and M-MDSC (CD11b^+^Ly6G^−^Ly6C^+^), in whole tumors from SHIP^WT^-HPC and SHIP^KO^-HPC mice. (**D**,**E**) Flow cytometry analysis and quantification of TAM subsets, M1-like TAM (CD11b^+^Ly6C^+^Ly6G^−^F4/80^+^CD206^−^CD86^+^) and M2-like TAM (CD11b^+^Ly6C^+^Ly6G^−^F4/80^+^CD86^−^CD206^+^), in the whole tumors of SHIP^WT^-HPC and SHIP^KO^-HPC mice. SHIP^WT^-HPC (WT-HPC) and SHIP^KO^-HPC (KO-HPC). Data are represented as the mean ± S.D. of SHIP^WT^-HPC (*n* = 3–4) and SHIP^KO^-HPC (*n* = 3–5) mice. * *p* < 0.05; ** *p* < 0.01 (by two-tailed *t* test).

**Figure 6 cancers-12-03631-f006:**
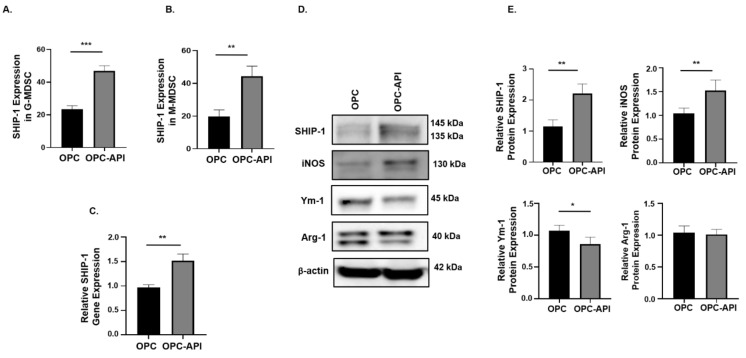
API Induces SHIP-1 Expression, and Skews TAM to an M1 Phenotype in OPC Mice. (**A**,**B**) Flow cytometry analysis of intracellular SHIP-1 in MDSC subsets, G-MDSC (CD11b^+^Ly6C^+/−^Ly6G^+^) and M-MDSC (CD11b^+^Ly6G^−^Ly6C^+^), from the pancreatic tumors of OPC and OPC-API mice. SHIP-1 expression was normalized to isotype CTRL. (**C**) Relative quantification of SHIP-1 gene expression in whole pancreatic tumors from OPC mice treated with vehicle and OPC mice treated with API. (**D**,**E**) Western blot analysis and quantification of normalized densitometry ratios of SHIP-1, iNOS, Ym-1 and Arg-1 protein in whole cell lysates of tumors from OPC and OPC-API mice. Appendix A shows the uncropped images. Data are represented as the mean ± S.D. of OPC (*n* = 3–6) and OPC-API (*n* = 4–6) mice. * *p* < 0.05; ** *p* < 0.01; *** *p* < 0.001 (by two-tailed *t* test).

**Figure 7 cancers-12-03631-f007:**
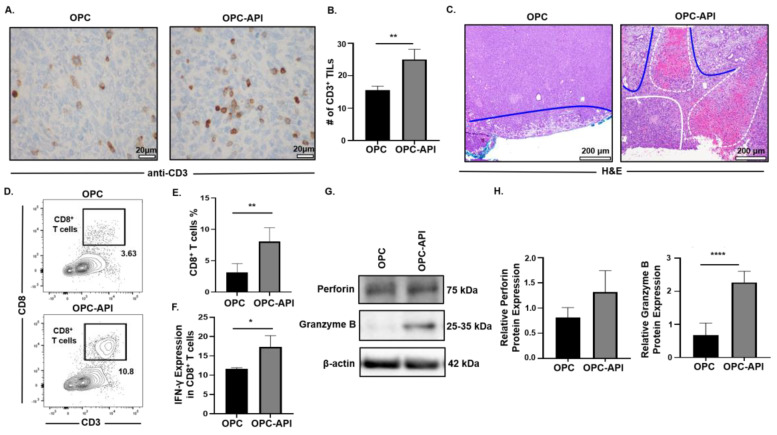
API Increases Anti-Tumor Immune Responses in the TME of OPC Mice. (**A**,**B**) Immunohistochemical images (600× magnification) and quantification of CD3^+^ TILs in the pancreatic tumors from OPC and OPC-API-treated mice stained with murine anti-CD3 antibody. (**C**) Histomorphology (H&E staining) (100× magnification) for pancreatic tumor necrosis (outlined in white dotted lines), viable stroma (outlined in blue lines), and viable pancreatic tumor (outlined in solid white line) from vehicle-treated OPC and OPC-API mice. (**D**,**E**) Flow cytometry analysis and representative quantification of CD8^+^ T cells (CD3^+^CD8^+^) in whole pancreatic tumors of vehicle-treated OPC mice and OPC-API mice. (**F**) Flow cytometry analysis and quantification of IFN-γ production in in vitro stimulated CD8^+^ T cells (CD3^+^CD8^+^) from the pancreatic tumor of OPC and OPC-API mice. (**G**,**H**) Western blot analysis and quantification of normalized densitometry ratios of Perforin and Granzyme B protein in whole lysates from OPC and OPC-API mice pancreatic tumors. Appendix A shows the uncropped images. Data are represented as the mean ± S.D. of OPC (*n* = 3–6) and OPC-API (*n* = 3–6) mice. * *p* < 0.05; ** *p* < 0.01; **** *p* < 0.0001 (by two-tailed *t* test).

**Figure 8 cancers-12-03631-f008:**
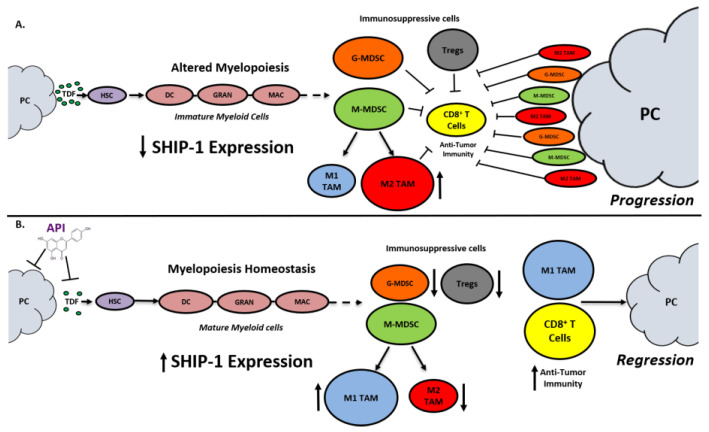
Proposed Model. Apigenin Increases SHIP-1 Expression, Augments Tumoricidal Macrophages and Improves Anti-Tumor Immune Responses in PC: (**A**) Pancreatic Tumor Microenvironment without Apigenin (API): Pancreatic cancer cells release tumor-derived factors (TDF) (green) that cause a decrease in SHIP-1 expression. The reduction in SHIP-1 expression causes hematopoietic stem cells (HSC) (purple) to skew towards altered myelopoiesis which yields immature myeloid cells (pink), including dendritic cells (DC), granulocytes (GRAN), and macrophages (MAC). This leads to the expansion of MDSC subsets, G-MDSC (orange) and M-MDSC (light green). M-MDSC mobilizes into the TME and differentiate into more pro-tumor M2-like TAM (red) compared to M1-like TAM (blue). MDSC subsets influence Treg (gray) expansion. MDSC subsets, M2-like TAM and Treg prevent anti-tumor immunity by blocking the migration of effector CD8^+^ T cells into the TME which leads to tumor progression. (**B**) Pancreatic Tumor Microenvironment with API: API targets and reduces the production of TDFs (directly or indirectly), leading to increased SHIP-1 expression, which promotes myelopoiesis homeostasis and the development of mature myeloid cells (DC, GRAN, and MAC). These M-MDSC that mobilize to the TME reprogram into tumoricidal M1-like TAM compared to M2-like TAM. Moreover, API treatment leads to a decrease in G-MDSC, M2-like TAM and Treg percentages in the TME. More importantly, API treatment increases the migration of effector CD8^+^ T cells (i.e., CD3^+^ TILs) into the TME, which elicits its anti-tumor activity, that causes pancreatic tumor regression.

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
