# Peer review of "Apigenin Increases SHIP-1 Expression, Promotes Tumoricidal Macrophages and Anti-Tumor Immune Responses in Murine Pancreatic Cancer"

_cancers, 2020, doi:10.3390/cancers12123631_

Round 1

Reviewer 1 Report

The authors have addressed the concerns of this reviewer.

Reviewer 2 Report

The manuscript submitted by Ghansah et al. highlights a promising therapeutic approach for pancreatic cancer using a bioflavonoid apigenin (API) in mouse model. SHIP1, a known activator of immune cells including myeloid cells and macrophages. Here, the authors elegantly demonstrated that API-induced SHIP1 increase could trigger an effective anti-tumor activity in pancreas. The results are very intriguing. Nevertheless, we would like to take privilege to comment on few aspects below.

  1. In Figure 1, the authors concluded that API exhibits anti-tumor activity with no apparent toxicity. However, since to tumor-associated inflammatory activity is not measured yet, therefore, we would rephrase it as, ‘API exhibits reduced tumor burden…’.

-We have made the suggested changes “API reduced tumor burden” in Line 115 and Figure 1.

Rev2: Accepted.

  1. We are also sceptical about the ‘toxicity’ part. What if API is administrated over a long time instead of 2-week study time frame which the authors demonstrated in the current study. Would a long term application be toxic for the animals? This needs to be thoroughly investigated before considering API treatment as a potential therapeutic approach.

-We previously published a survival study (i.e. approximately 40 days) using our heterotopic PC model treated with API (25mg/kg) that showed a delay in tumor progression, and an increase survival with no significant differences in weight of these mice at the end point of this study (PMID: 28152014). Moreover, the oral and IP LD50 values of Apigenin in mice are >5000 mg/kg and 3808 mg/kg, respectively (Osigwe CC et al. The Journal of Phytopharmacology 6: 38, 2017). In pharmacological studies 1/10th of LD50 value are used for repeated sub-chronic and chronic dosages. In this study, we used Apigenin IP dose of 25 mg/kg (3 times a week) which is 1/150th of LD50 value that is far less than 1/10th of LD50 value. Therefore, treatment of our PC models with API 25 mg/kg is well below toxicity levels in comparison to other pre-clinical studies. We have now addressed this point in the discussion section of the revised manuscript (Line 419-423).

Rev2: Explanation accepted.

  1. The scale is missing in Figure 1C. We request to provide it.

-Please note that Figure 1C does have a scale (gram) and it represents Pancreas/Tumor weights at the end of the study. However, if you were referring to Figure 1A, the Vevo ultrasound images, we have now resolved this issue and included the scales on the images in this revised manuscript.

Rev2: Apology from our side. We meant Figure 1A. We appreciate that authors payed attention to the details and added scale to Figure 1A.

  1. For Figure 1 C, we request the authors to show all the time points instead of showing the data from the end of the study. This would help visualizing how the tumor burden changes over the course of time for both OPC and OPC-API mice.

-Please note that Figure 1A and 1B (not Figure 1C) shows visual representation of in vivo tumor progression during the study represented in days (0-16) via Vevo ultrasound technology. Figure 1C represents actual tumor weights out of the animals at the end of study.

Rev2: Acceptable.

  1. We observed that the authors also performed imaging on spleens. We ask to measure the area and weight of the spleen for all time points as part of toxicity study. That way, they could capture any significant change occurring in spleen in OPC mice treated with API.

-Thank you for the suggestion. In response to Vevo imaging on the spleen, we were focused on tumor progression in the pancreas and did not focus on the spleen. We did not weigh the spleens at different timepoints. In our future experiments, we will measure spleen weight and area at different timepoints. In addition, we will need justification to secure approval through USF IACUC for this large-scale toxicity study with OPC mice.

Rev2: We thank the author for considering our concern. We highlighted the issue because, numerous promising anti-cancer agents nip in bud due the lack of proper toxicity test. As a result, FDA approval currently has a significant 97% failure rate at clinical trials for oncology (Wong et al.).

Reference:

Wong, C., Siah, K. and Lo, A. (2018). Corrigendum: Estimation of clinical trial success rates and related parameters. Biostatistics, 20(2), pp.273-286.

  1. As a comparable control, the authors should include a group of mice treated with API only. OPC-API group should be compared to this API only group in addition to the untreated control. Same is applicable for Figure 6.

-It is not ideal to use C57BL/6 mice (Tumor-Free) treated with API to compare MDSC and TAM subsets and western blot data in Figures 3, 4 and 6, which represents PC mice with TUMORS. However, we have now included data in the supplementary materials that shows C57BL/6 mice (tumor-free) treated with API compared to CTRL (Tx-vehicle) mice to evaluate MDSC subsets and CD8+ T cells in the spleen (Figure S4). Please note that there are no TAM subsets because there are no tumors in these CTRL in vivo experiments.

Rev2: Explanation accepted.

  1. If not, the in vivo experiment needs to be repeated twice to ensure reproducibility of the results.

-Yes. We have reproduced our in vivo data more than twice. Please review our updated statistical analysis section (Line 608-609). Our findings are consistent regarding percentages of MDSC and TAMs subsets from all our PC mice treated with vehicle and API.

Rev2: Thanks for including the information in the updated manuscript.

  1. In Figure 2C, the bands for SHIP1 do not look very convincing.

-We have repeated this experiment and generated a better figure of SHIP-1 protein expression which is now included in our revised manuscript (Figure 2C).

Rev2: We appreciate the authors’ effort to repeat the experiment.

  1. We request the author to provide better quality bands. A repeat of the procedure might avoid any possible artifacts during the Wester blot steps.

-We have repeated this experiment and generated a better figure of SHIP-1 protein expression which is now included in our revised manuscript (Figure 2C).

Rev2: OK

  1. The authors should also analyze SHIP1 at mRNA level by qPCR. This would strengthen the finding that API indeed induces SHIP1 expression, both at mRNA and protein levels.

-Yes, we have included SHIP-1 expression data via qPCR, WB and flow cytometry from our PC mice treated with vehicle and API in our original submission. Please look at manuscript figures and supplementary material (Figure 2C, 2D, 6A, 6B, 6C, 6D 6E, S1D, S1E, S1F, S2C, S7C, S7D, S7E).

Rev2: Accepted

  1. For Figure 3 D, the authors could combine percentage of both M1 and M2 TAMs in a single bar for each group to better visualize their shift between the treatment groups. Same applies for Figure 4D.

-Thank you for the suggestion. We have created a new graph of M1 and M2-like TAM percentages from OPC, KC-HPC, and SHIPWT-HPC and SHIPKO-HPC mice as suggested. These revised figures can be found in the results section of this manuscript (Figure 3, 4 and 5).

Rev2: Accepted

  1. We appreciate the authors’ effort in validating the data (e.g API Increases M1 TAM) using a second pancreatic cancer mouse model. However,

-Thank you for the comment.

  1. In Figure 4, the untreated control group is missing. We ask the authors to include it. They should also include an API only control group.

-Results in Figure 4 represent tumor data from a heterotopic PC model. CTRL mice would not have a tumor therefore, we cannot quantify MDSC and TAM subsets using flow cytometry. However, in the supplementary material we have included flow cytometry analysis of MDSC subsets from the spleen of CTRL (Tx-vehicle), KC-HPC (Tx-vehicle) and KC-HPC-API (Tx-API) (Figure S7). We have also addressed the API only treated CTRL group in comment 6.

Rev2: Reasonable. Accepted.

  1. n=3 is bare minimum. The experiment must be repeated using more mice per group.

-We have performed two independent experiments using the KC-HPC models. In these independent experiments, KC-HPC (Tx-vehicle) and KC-HPC (Tx-API) mice were 4 per group (n=4). Our flow cytometry findings from these repeated experiments using KC-HPC vs. KC-HPC-API mice are consistent regarding the percentages of MDSC and TAM subsets from the TME. Please review our updated statistical analysis section (Line 608-609).

Rev2: Thanks for the clarification.

  1. In Figure 2C, the authors used GAPDH as the housekeeping (HK) protein for Western blot whereas in Figure 6D, beta-actin was used. The discrepancy needs to be clarified. Although the function is same, yet for the sake of good lab practice (GLP) and data consistency/reproducibility, they should use either of the HKs but not both.

-We have reproduced our Western blot data in Figure 2C regarding the protein expression of β–actin (housekeeping protein) for consistency/reproducibility throughout the manuscript.

Rev2: We appreciate the effort.

  1. For Figure 7A and C, we request for better quality images with higher resolution. A scale should be included, too.

-Thank you for the suggestion. We have now provided our best quality images with high resolution for this manuscript. Also, Figure 7A and 7C now have scale bars as requested.

Rev2: Acceptable.

  1. Figure 7, the authors did not observe significant increase in perforin levels in OPC-API mice. We request to perform mRNA expression analysis for perforin.

-We now provide significant qPCR and immunofluorescence data of perforin and granzyme B expressions that can be found in the supplementary material section of the manuscript (Figure S12).

Rev2: Acceptable.

  1. As we know, CD8 T cells carry out cytotoxicity by perforin and granzyme B. Usually perforin makes the pores in cell membrane through which granzymes can get access within the tumor cell to carry out cytotoxic effects. Having said that, perforin alone can demonstarte cytotoxicity by destroying the cell membrane integrity.So, if the authors do not see increased level of perforin but increased level of granzyme B, this makes us skeptical about the mechanism of the anti-tumor activity. as granzyme B alone are unable to kill other cells. Therefore, we invite the authors to defend this issue.

-We agree with the reviewer’s point. We did see an increase trend of perforin protein expression which was not significant. However, we have now included qPCR and immunofluorescence data in the supplementary materials (Figure S12). We show a significant increase in perforin and granzyme B gene expressions in the TME of OPC-API vs. OPC mice (Figure S12). In addition, our immunofluorescence data shows a significant increase in perforin production (which allows granzyme B to enter and kill tumor) and an increase trend of granzyme B production in the TME of OPC-API vs. OPC mice (Figure S12).

Rev2: Thank you for reasoning and providing supporting data.

This manuscript is a resubmission of an earlier submission. The following is a list of the peer review reports and author responses from that submission.

Round 1

Reviewer 1 Report

Villalobos-Ayala et al studied the effects of the bioflavonoid apigenin (API) in heterotopic and orthotopic mouse models of pancreatic cancer (PC). In line with their previous studies showing that SHIP-1 acts as a tumor suppressor to inhibit immunosuppressive myeloid cells, the authors demonstrated that API reduced inflammation and increased SHIP-1 expression in PC mousemodels. They also showed that API treatment decreased MDSC expansion, skewed TAM to M1 phenotype, and increased tumor infiltrating T cells. Finally, the authors observed elevated tumor burden and pro-tumor macrophages in SHIP-1-deficient mice. They concluded that SHIP-1 is a potential therapeutic target, and the elevated SHIP-1 expression by API may provide a novel option for the treatment of PC. Compared to the known tumor cell-killing effects of API, the novelty of the study resides in the finding that API upregulates SHIP-1 and inhibits the functions of immunosuppressive myeloid cells in cancer models. The experiments were well designed and appropriately conducted. The results are important for understanding the anti-tumor activity of API. There are several concerns need to be addressed.

Major concerns:

  1. It was reported that API induces apoptosis and cell-cycle arrest, and inhibits DNA synthesis in human pancreatic cancer cell lines. The authors need to prove that the anti-tumor effect of API observed in this study was resulted from the anti-tumor immune response. Depletion of CD8 T cells in mice before API treatment may address this question.
  2. The authors may discuss the potential roles of SHP-1 and SHP-2 in PC development upon API treatment.

Minor concerns:

  1. The flow cytometry gating strategy: 1) There is no gating for living/dead cells. 2) The authors used CD11b+Ly6C+Ly6G-F4/80+CD206-CD86+for M1-like TAMs and CD11b+Ly6C+Ly6G-F4/80+CD86-CD206+for M2-like TAMs. However, it was reported that the Ly6C expression on TAMs is low or negative (PMID: 27325269, PMID: 27923825). In addition, there was no difference in CD86 levels in the gated M1 and M2 in this manuscript. Please clarify.
  2. There is discrepancy in the representative flow plots and quantification results in Figs 5B-C and 5D-E. The representative flow plots in Fig. 5B showed a significantly higher percentage of G-MDSC in SHIP KO mice vs WT mice. However, the quantification results in Fig. 5C showed no difference between these two groups.
  3. In Fig. 7B, how was the number of TILs determined? In Fig. 7D, the majority of cells were CD3+, which is different from Fig. 7A.

Reviewer 2 Report

The manuscript submitted by Ghansah et al. highlights a promising therapeutic approach for pancreatic cancer using a bioflavonoid apigenin (API) in mouse model. SHIP1, a known activator of immune cells including myeloid cells and macrophages. Here, the authors elegantly demonstated that API-induced SHIP1 increase could trigger an effective anti-tumor activity in pancreas. The results are very intriguing. Nevertheless, we would like to take priviledge to comment on few aspects below.

  1. In Figure 1, the authors concluded that API exhibits anti-tumor activity with no apparent toxicity. However, since to tumor-associated inflammatory acitvity is measure yet, therefore, we would rephrase it as, ‘API exhibits reduced tumor burden…’.
  2. We are also skeptical about the ‘toxicity’ part. What if API is administrated over a long time instead of 2-week study time frame which the authors demonstrated in the current study. Would a long term application be toxic for the animals? This needs to be thoroughly investigated before considering API treatment as a potential therapeutic approach.
  3. The scale is missing in Figure 1C. We request to provide it.
  4. For Figure 1 C, we request the authors to show all the time points instead of showing the data from the end of the study. This would help visualizing how the tumor burden changes over the course of time for both OPC and OPC-API mice.
  5. We observed that the authors also performed imaging on spleens. We ask to measure the area and weight of the spleen for all time points as part of toxicity study. That way, they could capture any significant change occurring in spleen in OPC mice treated with API.
  6. As a comparable control, the authors should include a group of mice treated with API only. OPC-API group should be compared to this API only group in addition to the untreated control. Same is applicable for Figure 6.
  7. If not, the in vivo experiment needs to be repeated twice to ensure reproducubility of the results.
  8. In Figure 2C, the bands for SHIP1 do not look very convincing. Therefore,
  9. We request the author to provide better quality bands. A repeat of the procedure might avoid any possible artifacts during the Wester blot steps.
  10. The authors should also analyze SHIP1 at mRNA level by qPCR. This would strengthen the finding that API indeed induces SHIP1 expression, both at mRNA and protein levels.
  11. For Figure 3 D, the authors could combine percentage of both M1 and M2 TAMs in a single bar for each group to better visualize their shift between the treatment groups. Same applies for Figure 4D.
  12. We appreciate the authors’ effort in validating the data (e.g API Increases M1 TAM) using a second pancreatic cancer mouse model. However,
  13. In Figure 4, the untreated control group is missing. We ask the authors to include it. They should also include an API only control group.
  14. n=3 is bare minimum. The experiment must be repeated using more mice per group.
  15. In Figure 2C, the authors used GAPDH as the housekeeping (HK) protein for Western blot whereas in Figure 6D, beta-actin was used. The discrepancy needs to be clarified. Altthough the function is same, yet for the sake of good lab practice (GLP) and data consistency/reproducibility, they should use either of the HKs but not both.
  16. For Figure 7A and C, we request for better quality images with higher resolution. A scale should be included, too.
  17. Figure 7, the authors did not observe significant increase in perforin levels in OPC-API mice. We request to perform mRNA expression analysis for perforin.
  18. As we know, CD8 T cells carry out cytotoxicity by perforin and granzyme B. Usually perforin makes the pores in cell membrane through which granzymes can get access within the tumor cell to carry out cytotoxic effects. Having said that, perforin alone can demonstarte cytotoxicity by destroying the cell membrane integrity.So, if the authors do not see increased level of perforin but increased level of granzyme B, this makes us skeptical about the mechanism of the anti-tumor activity. as granzyme B alone are unable to kill other cells. Therefore, we invite the authors to defend this issue.
